# E47 modulates hepatic glucocorticoid action

M. Charlotte Hemmer[1], Michael Wierer[2], Kristina Schachtrup[3], Michael Downes[4], Norbert Hübner[5], Ronald M. Evans[4] & N. Henriette Uhlenhaut [1,4,5,6]

Glucocorticoids (GCs) are effective drugs, but their clinical use is compromised by severe side effects including hyperglycemia, hyperlipidemia and obesity. They bind to the Glucocorticoid Receptor (GR), which acts as a transcription factor. The activation of metabolic genes by GR is thought to underlie these adverse effects. We identify the bHLH factor E47 as a modulator of GR target genes. Using mouse genetics, we find that E47 is required for the regulation of hepatic glucose and lipid metabolism by GR, and that loss of E47 prevents the development of hyperglycemia and hepatic steatosis in response to GCs. Here we show that E47 and GR co-occupy metabolic promoters and enhancers. E47 is needed for the efficient recruitment of GR and coregulators such as Mediator to chromatin. Altogether, our results illustrate how GR and E47 regulate hepatic metabolism, and might provide an entry point for novel therapies with reduced side effects.

[1] Molecular Endocrinology, Helmholtz Diabetes Center (HMGU) and German Center for Diabetes Research (DZD), IDO, Ingolstädter Landstr. 1, 85764 Neuherberg, Munich, Germany. [2] Department of Proteomics and Signal Transduction, Max Planck Institute for Biochemistry, Am Klopferspitz 18, 82152 Martinsried, Germany. [3] Center for Chronic Immunodeficiency, University Medical Center and Faculty of Biology, University of Freiburg, 79106 Freiburg, Germany. [4] The Salk Institute for Biological Studies & HHMI, 10010 N Torrey Pines Rd, La Jolla, CA 92037, USA. [5] Cardiovascular and Metabolic Sciences & DZHK (German Center for Cardiovascular Research), Charité-Universitätsmedizin & Berlin Institute of Health (BIH), Max Delbrück Center for Molecular Medicine in the Helmholtz Association (MDC), Robert-Rössle Strasse 10, 13125 Berlin, Germany. [6] Gene Center, Ludwig-Maximilians-Universität München (LMU), Feodor-Lynen-Straße 25, 81377 Munich, Germany. Correspondence and requests for materials should be addressed to N.H.U. (email: henriette.uhlenhaut@helmholtz-muenchen.de)

Glucocorticoids (GCs) are both widely used anti-inflammatory drugs and very potent metabolic regulators. Unfortunately, elevated GC levels are associated with metabolic disturbances like hyperglycemia, insulin resistance, dyslipidemia, hepatic steatosis and obesity. These symptoms are hallmarks of metabolic syndrome and compromise their long-term therapeutic use[1,2].

When GCs bind to the Glucocorticoid Receptor (GR), it translocates from the cytoplasm to the nucleus, where it regulates gene expression both positively and negatively. GR is a nuclear receptor known to bind to consensus DNA sequences termed glucocorticoid response elements (GREs), but the exact mechanisms leading to transcriptional activation versus repression are unclear[3–5]. In general, the desired immunosuppressant properties of GCs are thought to be due to the repression of inflammatory genes, while the adverse effects are believed to be caused by the activation of metabolic GR targets[6].

The past years have unveiled an extensive repertoire of interacting transcription factors and coregulators that affect gene regulation by GR. It has been shown that GR depends on the presence of lineage-determining pioneering factors to generate accessibility for enhancer and promoter binding and to create cell-type-specific hormone responses[5,7]. Indeed, GR cistromes are highly cell type specific (a cistrome is defined as the sum of all binding sites in a given cell type, essentially the entire ChIP-Seq data set). While GR is widely expressed, comparison of various cistromes from different cell types shows very little overlap. That means that the anti-inflammatory versus metabolic actions of GR might be encoded by both cell type specific accessibility to enhancers and tissue-specific crosstalk, which is in turn created by different pioneering or interacting factors. For example, GR binding at macrophage cis-regulatory elements occurs together with AP-1 and NF-κB at sites specified by Pu.1 and C/EBP, while the liver GR cistrome features co-occuring FoxA, HNF4α, HNF6 and C/EBP motifs[3,8,9].

To gain a deeper understanding of the mechanisms involved in the transcriptional activation of metabolic programs in response to GCs, we performed GR ChIP-Seq and motif analyses in mouse livers. In addition to the motifs described above, we found an E-Box specifically enriched near GREs in hepatic promoters and enhancers. This particular E-Box was predicted to be bound by the basic helix-loop-helix (bHLH) transcription factor E47. E47 is mostly known for its role in B and T cell lineage commitment, although it is expressed in many tissues[10,11]. It is encoded by the E2A or Tcf3 gene, and together with E12 (which arises from the same E2A gene by alternative splicing), E2–2 and HEB belongs to the class of E proteins that can heterodimerize with other bHLH factors. Furthermore, these E proteins are inhibited by binding to ID (Inhibitor of DNA binding) proteins, and this interplay is necessary to drive tissue and cell type specific gene expression programs. Importantly, mutation of all four ID proteins in mice has been linked to phenotypic alterations in glucose, lipid and cholesterol metabolism[12].

We therefore hypothesized that co-occupancy of GR and E47 might play a role for the transcription of a subset of genes and that E47 could modulate GR-dependent gene activation in a tissue-specific manner. Here we show that crosstalk between E47 and GR plays a role in hepatic lipid and glucose metabolism. Indeed, loss of E47 affects GR's ability to upregulate metabolic target genes. Consequently, E47 mutant mice are protected from steroid-induced hyperglycemia, dyslipidemia and hepatic steatosis. Using ChIP to map GR binding in mouse livers together with hepatocyte-specific E47 mutant mice, we demonstrate that GR and E47 synergize to mediate the metabolic actions of GCs at the genomic level. We find that inactivation of E47 leads to reduced occupancy of GR, Mediator and FoxO1 at a subset of hepatic enhancers and promoters. We confirm the relevance of these observations for human disease in a high throughput luciferase reporter screen of human cis-regulatory sequences responding to GCs. Thus, targeting this mechanism might provide an entry point for the development of GC therapies with reduced metabolic side effect profiles[13].

## Results

### GR and E2A co-occupy metabolic promoters and enhancers in vivo.

To gain mechanistic insight into GC-mediated gene regulation in metabolic tissue, we first performed ChIP-sequencing for GR in mouse livers collected in the late afternoon. This is the time when endogenous corticosterone levels rise in order to upregulate gluconeogenic genes in response to fasting[14,15]. As expected, we detected strong GR binding to the enhancers and promoters of known target genes such as Pck1, G6pc, Per1, TAT etc. (Fig. 1, Supplementary Fig. 1 & Supplementary Data 1). Bioinformatic motif analyses on GR-bound sequences similarly revealed consensus sites for transcription factors previously shown to co-occupy hepatic cis-regulatory elements, namely C/EBP, HNF4α, HNF6, FoxA, Stat5 and other nuclear receptors together with GREs[3,8]. Interestingly, we also found the E47 E-Box consensus motif CANNTG enriched near GREs (Fig. 1a). Of note, a previous study had also found an E47 motif associated with GR-bound fragments in mouse liver samples[16].

To validate that E47 plays a role for hepatic gene regulation, we first characterized its expression profile in mouse liver: Indeed, we detected robust levels of Tcf3 mRNA in livers collected every 4 h throughout the day/night cycle. We also found detectable levels of Heb, low levels of E2–2, high levels of Id2, robust levels of Id3 and Id1 and low expression of Id4 (Supplementary Fig. 1A). Measurable expression of the entire family of E and ID proteins suggests a functional role in hepatocytes, although none of them were differentially regulated in response to feeding/fasting, daily fluctuations in endogenous corticosterone secretion or GC treatment.

To test our hypothesis that E47 would bind together with GR to hepatic enhancers and promoters, we next performed ChIP-Seq using an E2A antibody in the same livers. As shown in Fig. 1b, we actually found that the E2A and the GR cistromes partially overlapped and that they shared a large number of binding sites. Significantly enriched motifs within E2A ChIP peaks included again C/EBP, HNF4α, FoxA, HNF6, E-Boxes and GREs, together with nuclear receptor DR1 and AP-1 motifs, underscoring the notion that E47 and GR share a set of common target genes (Fig. 1c). Interestingly, these common targets could be classified as genes involved in glucose and lipid metabolism. Functional annotation of GR and E2A co-bound cis-regulatory elements linked them to nearby genes important for glucose, lipid and fatty acid metabolism, such as those associated with metabolic syndrome, diabetes or hepatosteatosis (Fig. 1d). For example, GR and E2A were both bound to the promoters or enhancers of metabolic genes like glycerol-3-phosphate acyltransferase (Gpam), glucokinase (Gck), 24-dehydrocholesterol reductase (Dhcr24), 3-hydroxy-3-methylglutaryl-CoA synthase 1 (Hmgcs1), phosphoenolpyruvate carboxykinase 1 (Pck1), glucose-6-phosphatase (G6pc) and insulin like growth factor binding protein 1 (Igfbp1) (Fig. 1e and Supplementary Fig 1B). On the other hand, pathways that were specifically regulated by GR without E2A were for example small molecule metabolic processes, while genes involved in MAPK signaling showed only binding of E2A but not GR (Supplementary Fig. 1C, D).

Taken together, our ChIP-Seq experiments reveal that E2A co-occupies a subset of GR-bound enhancers and promoters in

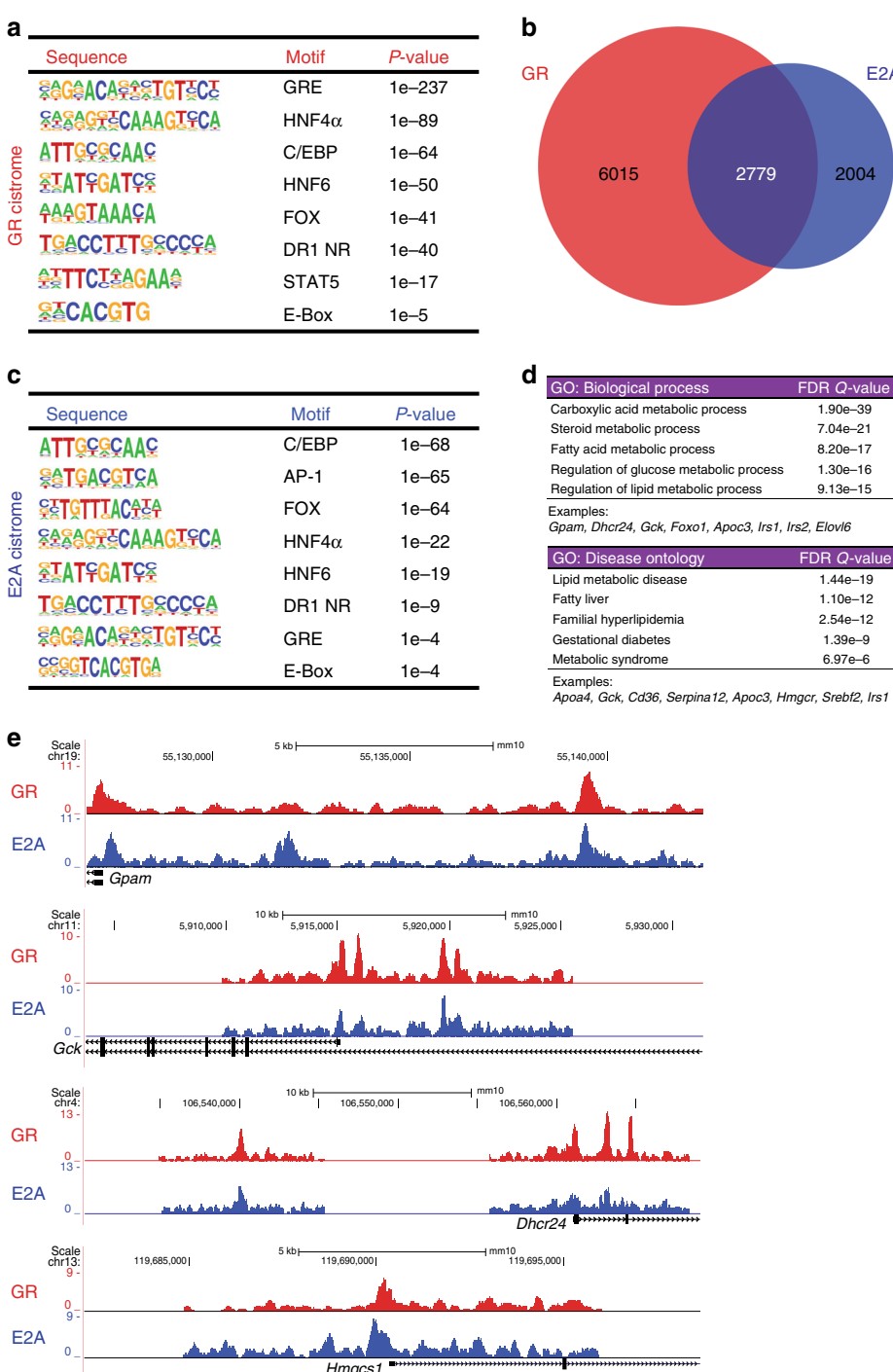

**Fig. 1** Genomic binding of GR and E2A overlaps in mouse liver. **a** Bioinformatic motif analyses of the hepatic GR cistrome detect enriched transcription factor consensus sites in GR ChIP sequences. **b** Area-proportional Venn diagram depicting the overlap between the GR and E2A liver cistromes. Numbers of called peaks in at least one sample are shown (*n* = 2 biol. replicates). **c** Analogous motif analyses of the hepatic E2A ChIP sequences. **d** Functional annotation of shared GR-E2A target genes by GO analysis, assigning common ChIP peaks to the nearest coding gene. **e** Representative ChIP-Seq tracks displaying co-occupancy of GR and E2A at metabolic promoters and enhancers. *Y*-axes show normalized read counts

mouse liver, at target genes governing hepatic glucose and lipid metabolism.

**E47 −/− mice are protected from side effects of GC treatment**. Since our ChIP-Seq experiments pointed towards a potential co-regulation of metabolic target genes by GR and E47, we studied the role of E47 for GR mediated transcription in vivo. Homo-zygous *E47* mutant mice lack mature B cells due to a failure to progress past the pro-B cell stage and have a partial early block in T-lymphocyte development[17]. Still, these mice are viable and have not yet been phenotypically characterized for metabolic functions. We therefore treated global *E47* knockout mice with exogenous GR ligands (Supplementary Fig. 2A). If E47 is important for the regulation of GR target genes, one would expect a reduced

transcriptional and phenotypic response to GCs upon loss of E47 function.

Similar to human patients developing so-called steroid diabetes, wild-type mice displayed mild hyperglycemia after 3 weeks of receiving Dexamethasone (Dex) in the drinking water[18]. When subjected to i.p. glucose tolerance testing, E47 −/− mice showed protection from this effect, with blood glucose levels being noticeably lower during the 90 min after the injection of the glucose bolus (Fig. 2a). Pyruvate tolerance tests performed in E47 knockouts also displayed lower values, suggesting reduced hepatic gluconeogenesis in these mice in response to Dex (Fig. 2b). In contrast, no difference in basal glucose or pyruvate tolerance was observed in untreated E47−/− animals (Supplementary Fig. 2B and C).

In line with these results, prominent downregulation of GR target genes involved in gluconeogenesis (Pck1, Gck, Igfbp1…) was observed in the livers of Dex-treated E47 knockout mice, which was consistent with GR's established function in the maintenance of blood glucose[15] (Fig. 2c).

We next used another protocol of chronic GC administration to model their effect on lipid metabolism:[19,20] After 3 weeks of corticosterone (Cort) treatment, wild-type littermates developed hepatic steatosis, while liver histology for E47−/− did not show any pronounced accumulation of lipid droplets (Fig. 2d). In addition, both liver and circulating triglycerides tended to be lower in E47 mutants in response to Cort treatment, but not in untreated mice (Fig. 2e and Supplementary Fig. 2D).

Accordingly, RNA-Seq performed on E47 mutant and wild-type livers after either Cort or Dex treatment showed a global deregulation of gene expression programs involved in lipid, triglyceride and cholesterol metabolism and transport. The reduced expression of these metabolic genes (such as apolipoproteins, CYP enzymes, cholesterol synthetases, Acetyl-CoA utilizing enzymes etc.) was confirmed by qRT-PCR (Fig. 2f, g, h and i and Supplementary Data 2). Importantly, the majority of genes affected by the loss of E47 were downregulated, as one would anticipate from diminished GR activity. Both GR and E47 can also act as transcriptional repressors, which might explain the upregulated transcripts. Furthermore, we measured very few gene expression changes in the livers of untreated E47−/− animals (Supplementary Fig. 2E), showing that this phenotype is specific to the GC response.

It should be noted that E47−/− mice had the same body weight and fat mass as controls (Supplementary Fig. 2F+G), but slightly elevated basal corticosterone levels, as could be explained by feedback on the hypothalamic-pituitary-adrenal (HPA) axis. When subjected to a Dex suppression test, both E47 −/− and controls suppressed endogenous corticosterone production, showing that their HPA axes were intact (Supplementary Fig. 2H). Despite published reports on the potential role for E47 in the regulation of adiponectin expression[21,22], we did not see differences in Adipoq mRNA levels in white adipose tissue from Dex-treated mutants (Supplementary Fig. 2I). Moreover, RNA-Seq expression profiles from E47−/− and WT quadriceps muscle and visceral adipose depots did not reveal striking differences in metabolic gene expression which would explain the observed differences in glucose tolerance and liver fat deposition (Supplementary Fig. 2J and K).

Interestingly, anti-inflammatory GC actions were retained in the absence of E47: Mutant bone marrow derived macrophages responded to Dex treatment with induction of the GR targets Per1 and Gilz, and repression of the inflammatory targets Ccl2 and Il6 just like controls (Supplementary Fig. 2L). This was not surprising, since E47 is not significantly expressed in macrophages, and we did not detect any E47 motifs enriched near GREs in the macrophage GR cistrome[9].

In summary, our data suggest that E47 knockout mice are protected from GC-induced hyperglycemia, dyslipidemia and hepatic steatosis due to a specific effect on metabolic gene expression programs in the liver.

**E47 modulates the GC response specifically in hepatocytes**. To confirm that the improved metabolic phenotype in response to GC treatment was indeed caused by impaired E47 function in hepatocytes, we generated liver-specific E47 mutants by crossing E47 floxed alleles with the Albumin-Cre line (Supplementary Fig. 3A)[23]. As shown in Fig. 3, glucose tolerance was significantly lower in Alb-Cre x E47^{flox/flox} (E47ΔLKO) mutants in response to Dex treatment (Fig. 3a, Supplementary Fig. 3B), and liver histology as well as liver and plasma triglyceride measurements showed protection from hepatic steatosis and dyslipidemia in Cort treated mice (Fig. 3b and c). Again, untreated liver-specific E47 mutants did not display any significant differences to wild type with respect to basal glucose tolerance, body weight or fat mass gained on Cort (Supplementary Fig. 3C, D and E).

Concordantly, these phenotypes were again coincident with the differential expression of metabolic gene programs governing glucose, lipid, fatty acid and cholesterol utilization (like Pck1, Acacb, Dhcr7 etc.) as measured by RNA-Seq (Fig. 3d and Supplementary Data 2) and qRT-PCR in livers of Dex or Cort treated mice (Fig. 3e and f). Of note, the RNA-Seq experiment shows acute responses after only 1 h of Dex injection to avoid indirect effects due to disease progression or pathological changes.

Importantly, a substantial fraction (63–80%) of the transcripts differentially expressed in E47 mutant mice treated with GCs (Figs. 2 and 3) could be classified as direct GR target genes, as defined by a nearby GR ChIP-Seq peak (Fig. 3g and Supplementary Fig. 3F). This underscores our interpretation that the loss of E47 affects the transcriptional regulation of GR-bound loci.

In conclusion, our data demonstrate that E47 is required specifically in hepatocytes for the activation of gluconeogenesis, lipid and cholesterol storage and triglyceride synthesis, and that targeting liver E47 improves metabolic parameters in response to GCs.

**Efficient regulation of metabolic genes by GR requires E47**. Since genetic loss of E47 revealed its role in the activation of hepatic glucose and lipid metabolism by GR, we tested whether impaired function of the receptor itself would explain its reduced capacity. However, both total GR mRNA and protein levels were unaffected by the loss of E47, and GR nuclear localization was unchanged in mutant mouse livers (Supplementary Fig. 4A, B and C).

We therefore performed Co-IP and ChIP-MS experiments to study protein protein interactions between GR and E47 inside their shared transcriptional complex. The E47 protein sequence contains two predicted LXXLL nuclear receptor interaction motifs. Indeed, we detected a weak physical interaction between GR and E2A by Western Blot from endogenous liver IPs (Supplementary Fig. 4D). Furthermore, we purified hepatic co-regulators from chromatin cross-linked to GR by proteomics in both wild type and E47 mutant livers treated with Dex (Fig. 4a +b). We found that C/EBPs, SRCs, HNF4α, RXR, FoxO1, Mediator subunits, chromatin remodelers and histone modifiers were enriched together with GR. In E47 mutant livers, however, Med16, Med23 and FoxO1 peptides were not present in the ChIP-MS data set. (The only other proteins that were significantly enriched in WT samples but not in E47ΔLKO were Elongation factor Tu Tufm, Epithelial splicing regulatory protein 2 Esrp2, YTH domain-containing protein 1 Ythdc1, Heterogeneous

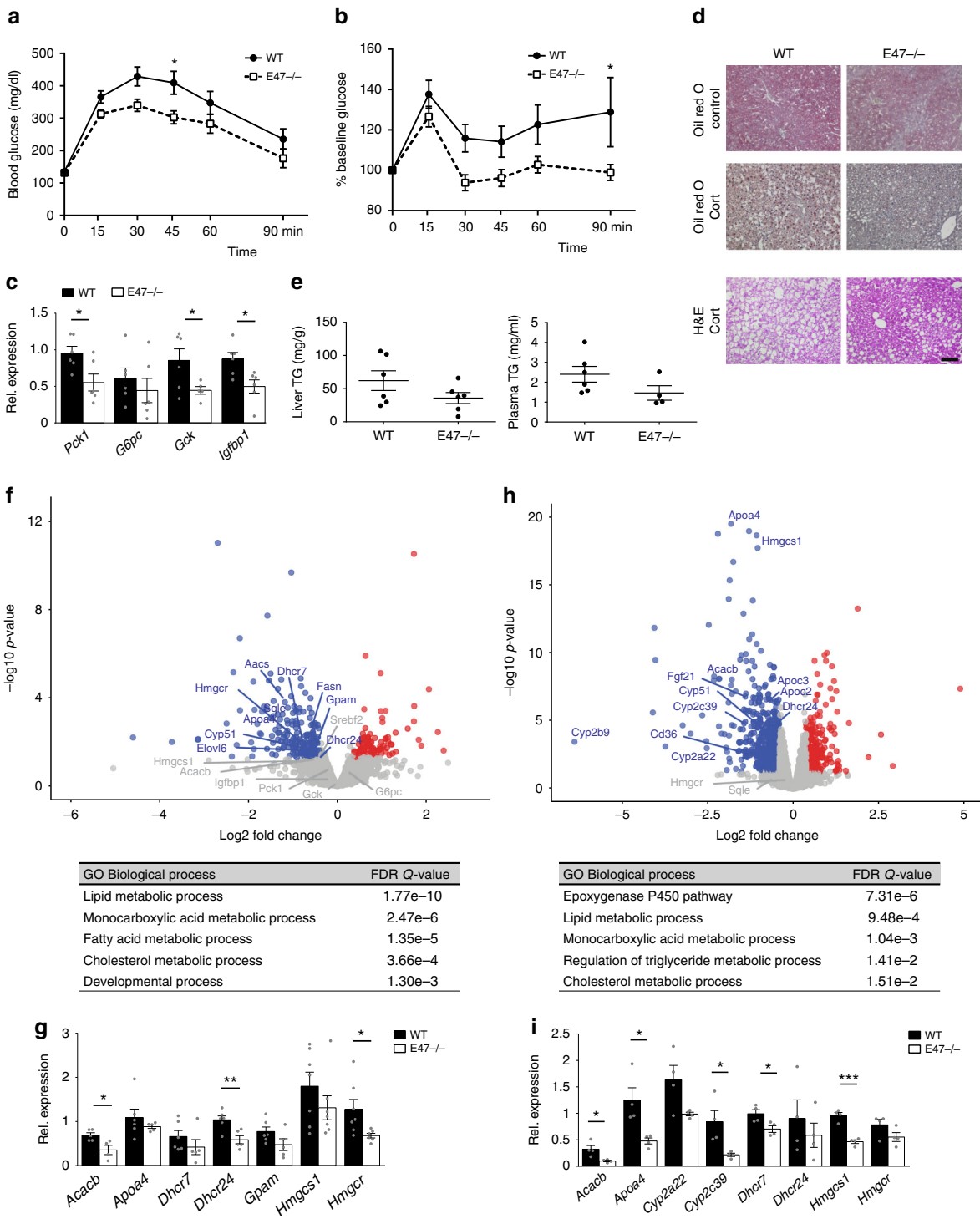

**Fig. 2** *E47* knockout mice are protected from GC-induced side effects. **a** i.p. glucose tolerance test (GTT) and **b** i.p. pyruvate tolerance test (PTT) of E47−/− and WT mice on Dex. Data were analyzed by ANOVA and Bonferroni's multiple comparison test and are shown as mean ± SEM. Asterisks indicate significance, *$P < 0.05$, $n = 10$ per genotype (GTT); $n = 7$ (WT) & 9 (E47−/−) (PTT). **c** qRT-PCR of gluconeogenic genes in Dex-treated livers, normalized to *U36b4;* data are mean ± SEM *$P < 0.05$, Student's *t* test. $n = 6$ (WT) & 5–6 (E47−/−). **d** Liver sections stained with Oil Red O and hematoxylin and eosin (H&E) after Cort or vehicle treatment. ×20 magnification; black scale bar: 100 μm, representative image from $n = 3$. **e** Liver and plasma triglycerides on Cort. Data are mean ± SEM, $n = 6$ (WT) and 4–6 (E47−/−). **f** Volcano plot and GO analysis of deregulated genes (blue = downregulated; red = upregulated, FC 1.3, $P < 0.05$) in livers of E47−/− mice on Dex, $n = 2$ (WT) & 3 (E47−/−). For GO analysis a base mean cutoff > 200 was used. **g** qRT-PCR of metabolic genes on Dex, normalized to *U36B4*. Data are mean ± SEM, *$P < 0.05$, Student's *t* test, $n = 5$–7 (WT) & 4-7 (E47−/−). **h** Volcano plot and GO analysis of deregulated genes (blue = downregulated; red = upregulated, FC 1.3, $P < 0.05$) in livers of E47−/− mice on Cort, $n = 4$ per genotype. For GO analysis a base mean cutoff > 200 was used. **i** qRT-PCR of differentially expressed genes on Cort, normalized to *U36b4;* data are mean ± SEM *$P < 0.05$, Student's *t* test. $n = 4$ per genotype

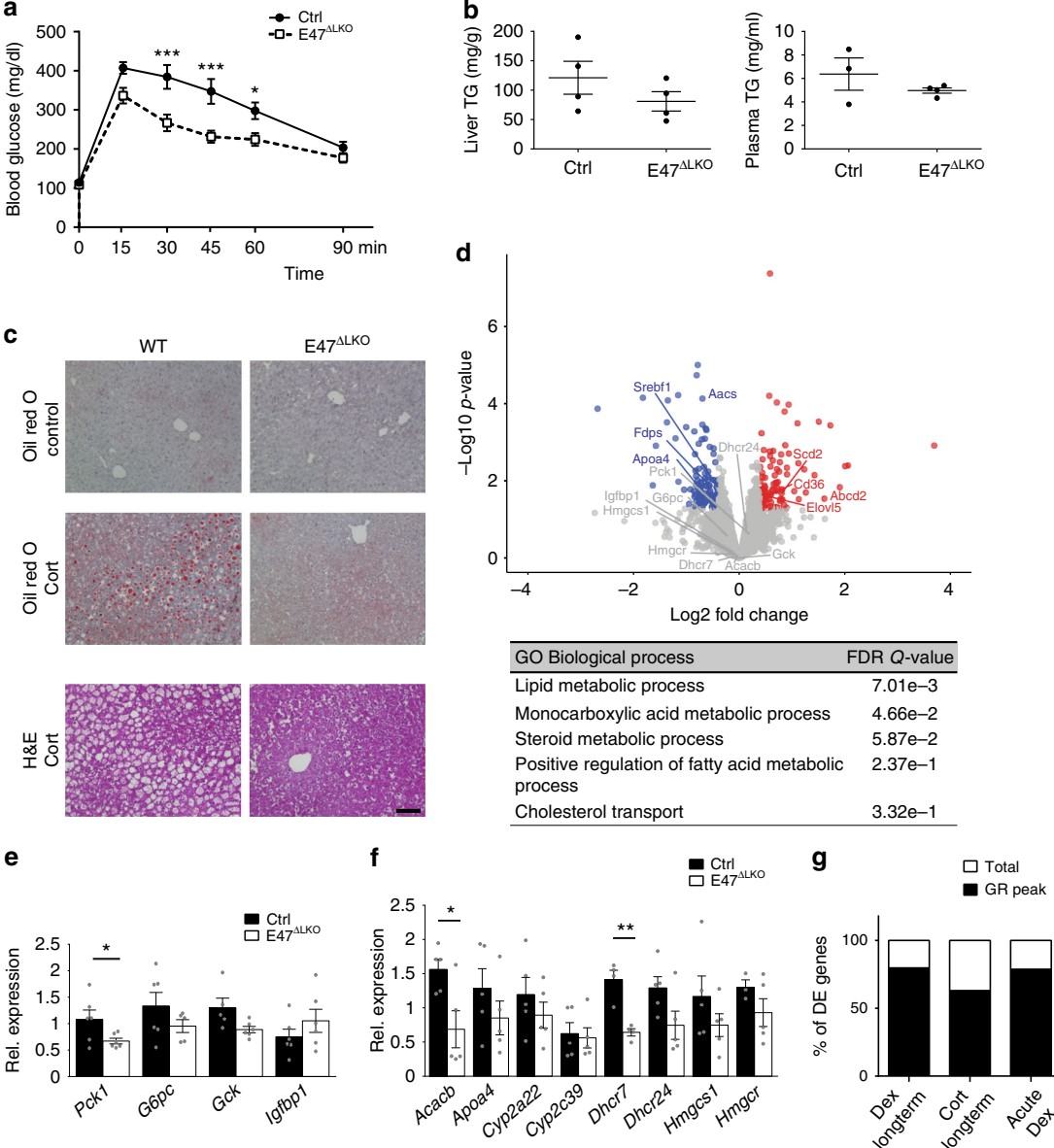

**Fig. 3** Hepatocyte-specific deletion of *E47* recapitulates the protective effect. **a** i.p. GTT in *Albumin-Cre x E47^flox/flox* (E47^ΔLKO) and control mice on Dex. The data were analyzed by ANOVA and Bonferroni's multiple comparison test and are shown as mean ± SEM. Asterisks indicate significance, *$P < 0.05$, ***$P < 0.001$, $n = 11$. **b** Liver and plasma triglycerides after Cort. Data are mean ± SEM, $n = 4$ per genotype. **c** Liver sections stained with Oil Red O and H&E before and after Cort. ×20 magnification, black scale bar: 100 μm, $n = 1$. **d** Volcano plot and GO analysis of deregulated genes (blue = down-; red = upregulated, FC 1.3, $P < 0.05$) in E47^ΔLKO livers after 1 h Dex ($n = 3$). For GO analysis a base mean cutoff > 200 was used. **e** qRT-PCR of gluconeogenic genes in livers on Dex, normalized to *U36b4*. The data are mean ± SEM, *$P < 0.05$, Student's $t$ test, $n = 6$. **f** qRT-PCR of metabolic genes after Cort, normalized to *U36b4*. The data are mean ± SEM, *$P < 0.05$, Student's $t$ test, $n = 5$. **g** Percentage of genes differentially expressed in *E47* mutant mice (Fig. 2 and 3) which harbor a detectable GR ChIP peak nearby

nuclear ribonucleoprotein L Hnrnpl and the Ribosomal protein L15 Mrpl15).

Med23, Med24 and Med16 are components of the Mediator complex which acts as a bridge between specific transcription factors and the general RNA Polymerase II transcription machinery[24]. Its core component Med1 was previously shown to be required for target gene regulation by GR[25]. FoxO1 is a direct transcriptional regulator of hepatic lipid metabolism and gluconeogenesis downstream of insulin signaling[26]. Thus, we hypothesized that close binding of GR and E47 at certain loci is required to assemble a critical mass of transcriptional regulators for efficient activation of these genes. We would like to point out that we did not measure reduced mRNA expression of either

Mediator components or FoxO1 itself in *E47* mutant tissue (Supplementary Fig. 4A).

For that reason, we performed ChIP-qPCR studies for GR, Med1 and FoxO1 on hepatic *cis*-regulatory elements in both WT and *E47* mutant livers. As shown in Fig. 4c, we found that loss of E47 led to reduced recruitment of GR itself at certain metabolic loci (such as *Apoa4*, *Dhcr24*, *Gpam*, *Hmgcr* or *Pck1*), consistent with E47's known function as a lineage determining pioneer factor providing chromatin accessibility[27]. We also found diminished occupancy of Med1 and potentially FoxO1 at some of these sites, for example, the *Gpam* enhancer, the *Apoa4* promoter and the *Pck1* promoter (but not at the *Acacb* locus). These data suggest that E47 is necessary for binding of GR,

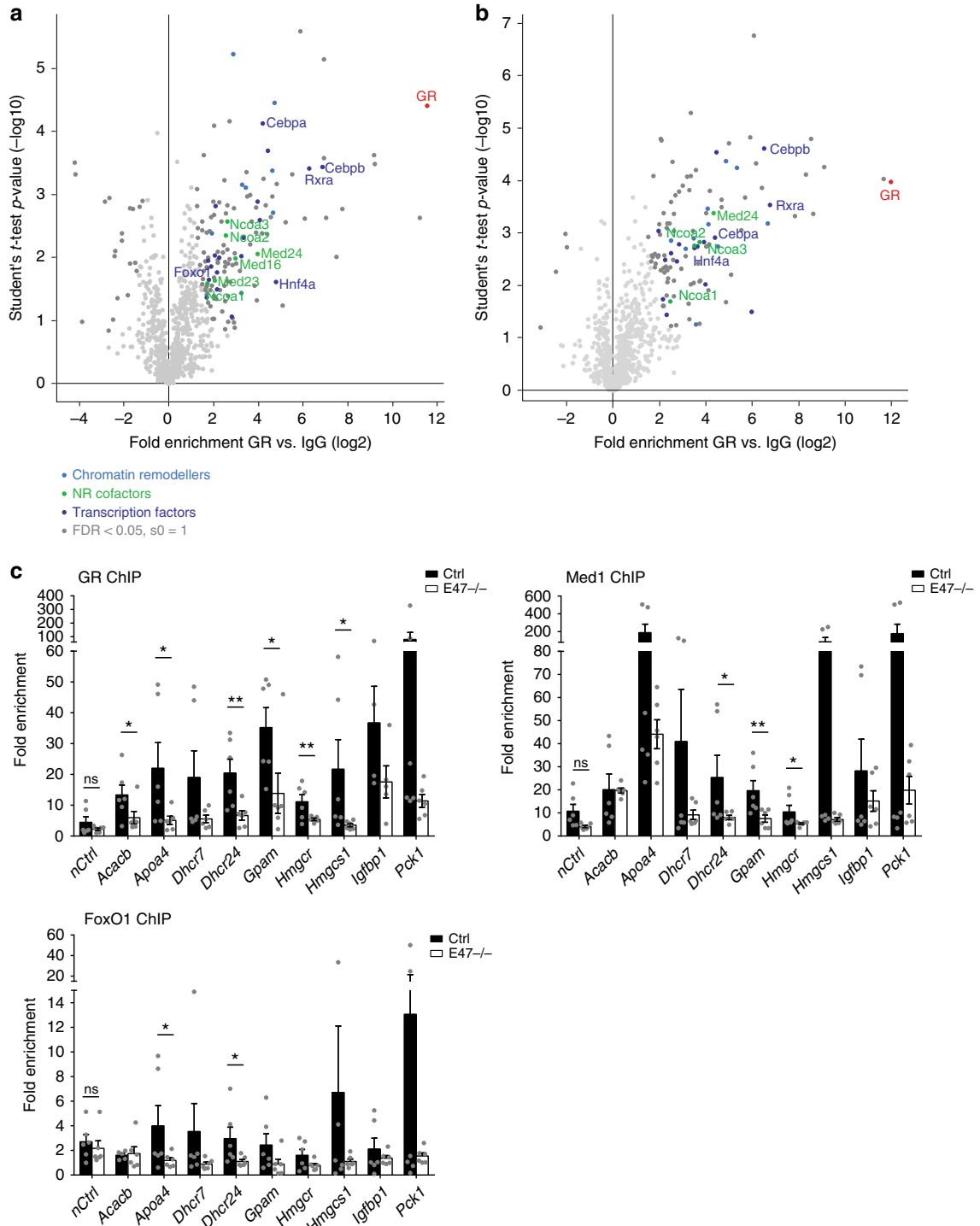

**Fig. 4** E47 is required for efficient chromatin binding of GR, FoxO1 and Mediator. **a** ChIP-MS for GR in Dex-injected control livers. Pale blue: chromatin remodelers (GO Biol. Process); green: NR coregulators; dark blue: sequence specific DNA binding transcription factor activity (GO Mol. Function); dark grey: significant outliers derived by Fisher's exact test (FDR < 0.05, s0 = 1).; $n = 3$ biological replicates. **b** ChIP-MS for GR in Dex-injected E47$^{\Delta LKO}$ livers. Pale blue: chromatin remodelers (GO Biol. Process); green: NR coregulators; dark blue: sequence specific DNA binding transcription factor activity (GO Mol. Function); dark grey: significant outliers derived by Fisher's exact test (FDR < 0.05, s0 = 1).; $n = 3$ biological replicates. **c** ChIP-qPCR in Dex-treated E47−/− and wild-type livers shows binding of GR, Med1 and FoxO1 at metabolic promoters and enhancers. The data are shown as fold enrichment over IgG and normalized to a negative control region. Data are mean ± SEM, *$P < 0.05$, **$P < 0.01$ Student's $t$ test, $n = 6$, ns not significant

Mediator and possibly FoxO1 to the promoters and enhancers of certain common target genes, and for their efficient induction by GCs (Supplementary Fig. 4E).

Taken together, we find that co-occupancy of GR and E47 on a subset of liver enhancers is required for effective loading of the

Mediator complex and transcriptional activation of metabolic gene expression by GR and FoxO1 in response to GCs.

**E47 affects the physiological response to fasting.** Endogenously, an important function of GCs is the maintenance of blood

glucose during times of fasting, when hormone levels are elevated. To test whether our observed coregulation of metabolic programs by GR and E47 also contributes to the physiological response to fasting, we withdrew food from both total and liver-specific *E47* mutants and controls for up to 48 h. Under these conditions, liver-specific *Alfp-Cre GR^flox/flox* mice were reported to become hypoglycemic[15]. As shown in Fig. 5a, elevation of GC levels upon fasting similarly led to reduced occupancy of GR and Med1 at the *Apoa4, Dhcr24, Gck, Gpam, Pck1* etc. *cis*-regulatory elements in the absence of E47 (see also Supplementary Fig. 5A, B, C). This was accompanied by a slight reduction in the expression levels of common metabolic target genes such as *Acacb, Apoa4* and *Pck1* (Fig. 5b). Despite this reduced GR activity at the genomic level, neither global nor liver-specific *E47* mutants displayed fasting hypoglycemia (Fig. 5c and Supplementary Fig. 5D). The mice show a tendency towards lower serum glucose levels after 24 h of fasting, but their reduction in hepatic gluconeogenesis is most likely compensated by the kidney, as has been shown for GR itself[28]. However, loss of E47 resulted in lower hepatic triglyceride contents in fasted mice (Fig. 5d and Supplementary Fig. 5E), which together with reduced metabolic gene expression points to a role for E47 in the physiological adaptation to starvation.

In conclusion, E47 appears to be involved in adequate gene activation of GR target genes during the hepatic response to fasting, which mainly affects triglyceride storage and utilization rather than circulating glucose.

**Human GR targets are co-regulated by E47.** To test whether this E47 mediated metabolic response to GCs is conserved between mice and humans, we screened a set of 162 human *cis*-regulatory sequences predicted to be regulated by GR in a high throughput luciferase reporter assay. In this assay, cells were transiently transfected with commercially available GR-responsive reporter constructs in medium supplemented with either Cortisol or Dexamethasone (or vehicle). Luciferase activity was measured and normalized, and individual reporter sequences were classified into those mediating either activation or repression (or not responding) by GCs (Fig. 6a; Supplementary Data 3). For example, around 40 human reporters were downregulated, while more than 80 were induced, at least twofold by GCs. As expected, classical GRE consensus sequences were found enriched in both positive and negative human *cis*-regulatory elements. Interestingly, around half (47%) of all activated reporter sequences featured an identifiable E47 motif, which means E47 consensus sites were significantly enriched near GREs in human promoters or enhancers activated by GR (Fig. 6b).

Co-transfecting these reporters with expression vectors for either human GR or E47 alone, or GR together with E47, indeed showed induction of luciferase activity only when both transcription factors were present (Fig. 6c). Mutation of the predicted E-Box motifs, for example in the *ATP2B3* promoter, prevented regulation by E47 and abrogated this positive effect (Fig. 6d). Furthermore, co-transfection with an expression construct for the human inhibitor ID3 also abolished this cooperative induction, as shown here for the *DPEP1* promoter (Fig. 6e). These results show that E47 binds to and activates E-Boxes in human promoters or enhancers together with GR.

Taken together, our luciferase reporter data set demonstrates a conserved function for E47 in the transcriptional activation of certain human gene programs by GCs.

## Discussion
GCs are a linchpin of anti-inflammatory therapy, but their utility is eventually limited by considerable side effects, including hyperglycemia, obesity, dyslipidemia or hypertension[29].

Separating the beneficial immune suppressive actions from their adverse metabolic effects is the ultimate goal for the development of safer drugs[30]. In general, downregulation of GR target genes is assumed to be crucial for the anti-inflammatory actions, while upregulation is thought to underpin the undesirable effects. Unfortunately, compounds such as SEGRAMs (selective GR agonists and modulators), which favor the repressive GR functions over the activating potential, have not been very successful[6].

We have identified E47 as a transcription factor cooperating with GR specifically in the transcriptional activation of metabolic gene networks in the liver. Consequently, *E47* mutant mice are protected from GC-induced hyperglycemia, hyperlipidemia and hepatic steatosis. We could show that the presence of E47 is required for efficient GR binding at a subset of liver-specific metabolic promoters and enhancers. Interestingly, loss of E47 also affects the recruitment of the Mediator complex and FoxO1 (Fig. 7).

These observations agree with E2A's known role as lineage determining pioneering factor. For example, E47 has been shown to affect Pu.1 occupancy during B cell maturation[27]. Previous genomic studies in lymphoid cells have also hinted at a possibility of E47 regulating genes involved in lipid metabolism[31]. In addition, E47 has been linked to FoxO1 in a common pathway, albeit in B cells, with E47 being upstream of FoxO1[32,33]. We have now uncovered a novel role for E47 in modulating nuclear receptor and potentially FoxO1 function in the liver. All FoxO1/3/4 factors play vital roles in mammalian glucose homeostasis. FoxO1 is a prominent regulator of hepatic gluconeogenesis in response to insulin signaling[34]. Similar to our findings, liver-specific *FoxO1/3/4* mutant mice also display lower blood glucose levels and improved glucose and pyruvate tolerance[35]. Moreover, FoxO1 was recently shown to synergize with GR in hepatic glucose production and lipogenesis, for example in response to both insulin and Dex on the *Gck* promoter[36]. The FOX consensus motifs that we identified in the GR and E2A cistromes are mainly co-occupied by GR and the FoxA factors[8], but could of course be recognized equally well by FoxO1.

Mediator is a large complex that is generally required for transcription by RNA Polymerase II. It serves as a central scaffold that relays signals from transcription factors bound to enhancers (such as GR) to the transcription initiation machinery assembled at promoters[37]. Importantly, liver-specific *Med1* mutant mice are also protected from GC-induced hepatic steatosis due to downregulation of GR target genes responsible for glucose and lipid metabolism[38]. Further agreeing with our findings, liver-specific deletion of *Med23* has been reported to improve lipid profiles and glucose tolerance by affecting the transcriptional activity of FoxO1[39]. Reduced binding and transcriptional activity of GR, FoxO1 and Mediator in the absence of E47 therefore presents a likely molecular mechanism underlying our phenotype.

One promising case of a tissue-selective modulator of GC signaling appears to be treatment with an LXRβ antagonist: Interestingly, mice treated with GSK2033, as well as *LXRβ* knockout mice, are also spared from the diabetogenic effects of exogenous GC administration. Mirroring our results, Patel et al. report reduced recruitment of Med1 and GR itself to the *Pck1* promoter in these mice[40,41]. We did not see lower expression of either *LXRα* or *LXRβ* mRNA in *E47* mutant livers (Supplementary Fig. 4A), but we identified nuclear receptor DR1 motifs in both the GR and E2A cistromes. Since the LXR heterodimer partner RXR was co-purified with GR in our proteomics experiment, it is conceivable that GR, E47 and LXRβ share a common set of overlapping metabolic target genes, such as *Pck1*. Loss of either LXRβ or E47 might therefore exert its beneficial effects via a similar mechanism.

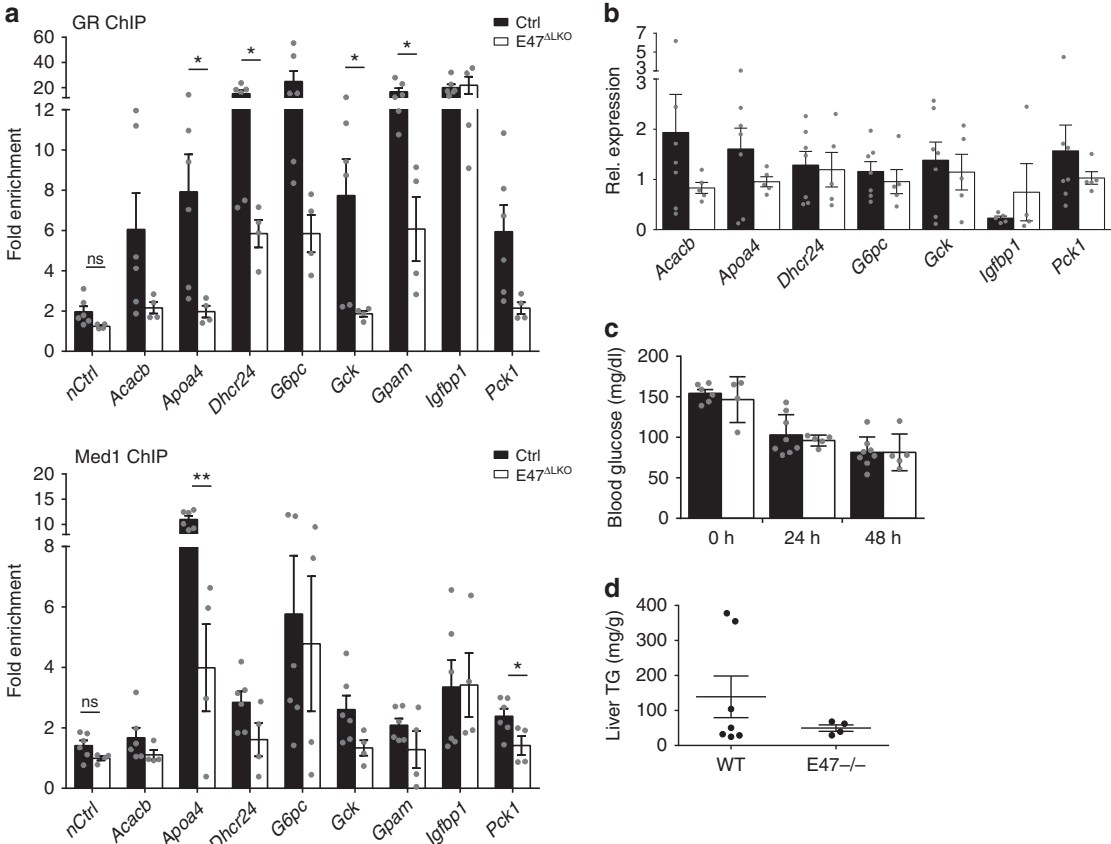

**Fig. 5** E47 is partially involved in the hepatic response to fasting. **a** ChIP-qPCR in fasted $E47^{\Delta LKO}$ and control livers shows binding of GR and Med1 at metabolic promoters/enhancers. The data are fold enrichment over IgG, $n = 6$ (Ctrl) and 4 ($E47^{\Delta LKO}$). The data are mean ± SEM, *$P < 0.05$, **$P < 0.01$, Student's $t$ test. **b** qRT-PCR of metabolic genes in fasted livers, normalized to $U36b4$. ($Gpam$ transcripts were very low or undetectable in this experiment.) Data are mean ± SEM, $n = 5$–7 (WT) & 4–5 ($E47-/-$). **c** Blood glucose levels of $E47-/-$ mice and controls measured at 0 h, 24 h and 48 h of fasting. The data are mean ± SEM, $n = 6$–8 (WT) & 4–5 ($E47-/-$). **d** Liver triglycerides in 48h-fasted $E47-/-$ mice and controls. The data are mean ± SEM, $n = 7$ (WT) & 4 ($E47-/-$)

Taken together, we have identified a novel role for E47 in metabolic gene regulation in hepatocytes in response to GCs. We have not only shown that targeting E47 might be a novel approach to eliminate certain side effects of GC treatment, while maintaining their potent anti-inflammatory properties, but present a case of cell-type and locus-specific gene regulation by GR. E47 only shares a subset of target genes with GR and therefore modulates the GC response in a specific manner. Since this function is conserved on several human promoters and enhancers, one might speculate that the presence, absence or the difference in E47 expression levels might influence patients' sensitivity to GCs in several different scenarios. Finally, this shared GR- and E47-bound, deregulated target gene set which is responsible for the protective effect, might present a further entry point for the development of alternative therapeutic strategies.

## Methods

**Animals**. $E47$ knockout and $E47$ floxed alleles were bred on a C57BL/6 background[17]. For $E47$ knockout mice, wild-type littermates served as controls. E47 floxed mice were crossed with hepatocyte-specific Albumin (Alb)-Cre mice obtained from JAX (B6.Cg-Tg(Alb-cre)21Mgn/J). Alb-Cre negative floxed littermates served as controls. Mice were housed in a controlled SPF facility with a 12 h dark/light cycle at an ambient temperature of 23 °C with constant humidity and fed ad libitum with regulator chow diet (Altromin GmbH). For all experiments, 10- to 16-week-old males were used. Formal ethical approval for animal experiments was obtained from the relevant authorities (LAGeSo Berlin, Reg 0103/11; district government of Upper Bavaria 2532–158–2014) in accordance with MDC & HMGU guidelines for the care and use of animals.

**Chromatin Immunoprecipitation (ChIP)**. ChIP was performed as previously described:[9] Mouse livers were harvested, snap-frozen and homogenized in lysis buffer (10 mM Hepes-KOH, 10 mM KCl, 5 mM MgCl$_2$, 0.5 mM DTT) using the TissueLyser (Qiagen) with steel beads. Lysates were passed through a cell strainer and cross-linked in 1% formaldehyde for 15 min and quenched with 0.2 M glycine for 5 min. Chromatin was sonicated to a 0.1–1 kb size using a Bioruptor (Diagenode) and incubation with the following antibodies was set up overnight: rabbit αGR (#24050–1-AP, Proteintech), rabbit αE2A (SC-349 ×, Santa Cruz Biotechnology), rabbit IgG (#2729, Cell Signaling), rabbit αMed1 (#A300–793A, Bethyl laboratories) and rabbit αFoxo1a (#ab39670, Abcam). For ChIP-Sequencing, 8 µg of each antibody was used. For ChIP qPCR, 3 µg of each antibody was used. The chromatin complexes were immunoprecipiated using Dynabeads M-280 (Invitrogen). ChIP DNA was purified and quantified using the Power SYBR Green Master Mix (Life Technologies) in a ViiA 7 Real-Time PCR System (Thermo Fischer Scientific). Primers for ChIP qPCR are listed in Supplementary Table 1a.

**ChIP-sequencing and data analysis**. Libraries were prepared using the KAPA Hyperprep Kit (Kapa Biosystems, KK8504). Illumina compatible adapters were synthesized by IDT (Integrated DNA Technologies) and used at a final concentration of 68 nM. Size-selection (360–610 bp) of adapter-ligated libraries was performed using 2% dye free gels (#CDF2010, Sage Science) in a Pippin Gel station (Sage Science). qPCR was used to estimate library concentrations with the KAPA Library Quantification Kit (#KK4873, Kapa Biosystems). Library quality was verified using the Agilent High Sensitivity DNA Kit (Agilent) in a 2100 Bioanalyzer (Agilent).

Libraries were subjected to NGS on an Illumina HiSeq4000. Reads were aligned to the mouse mm10 reference genome using BWA-MEM version 0.7.13[42] and duplicates were removed using Picard Tools version 2.8.3 (http://picard.sourceforge.net/). Reads were filtered for uniquely mapped read pairs with samtools[43] and downsampled to 18 mio (GR ChIP-Seq) or 22 mio (E2A ChIP-Seq) read pairs (see https://github.com/FranziG/E47KO-liver). To visualize the tracks, mapped reads were converted to bedGraph using HOMER version 4.9[27], filtered for called peak regions ± 2 kb and displayed on the UCSC genome browser. Peaks were called using MACS2 version 2.1.1.20160309[44] and Gene Ontology analysis performed with GREAT[45].

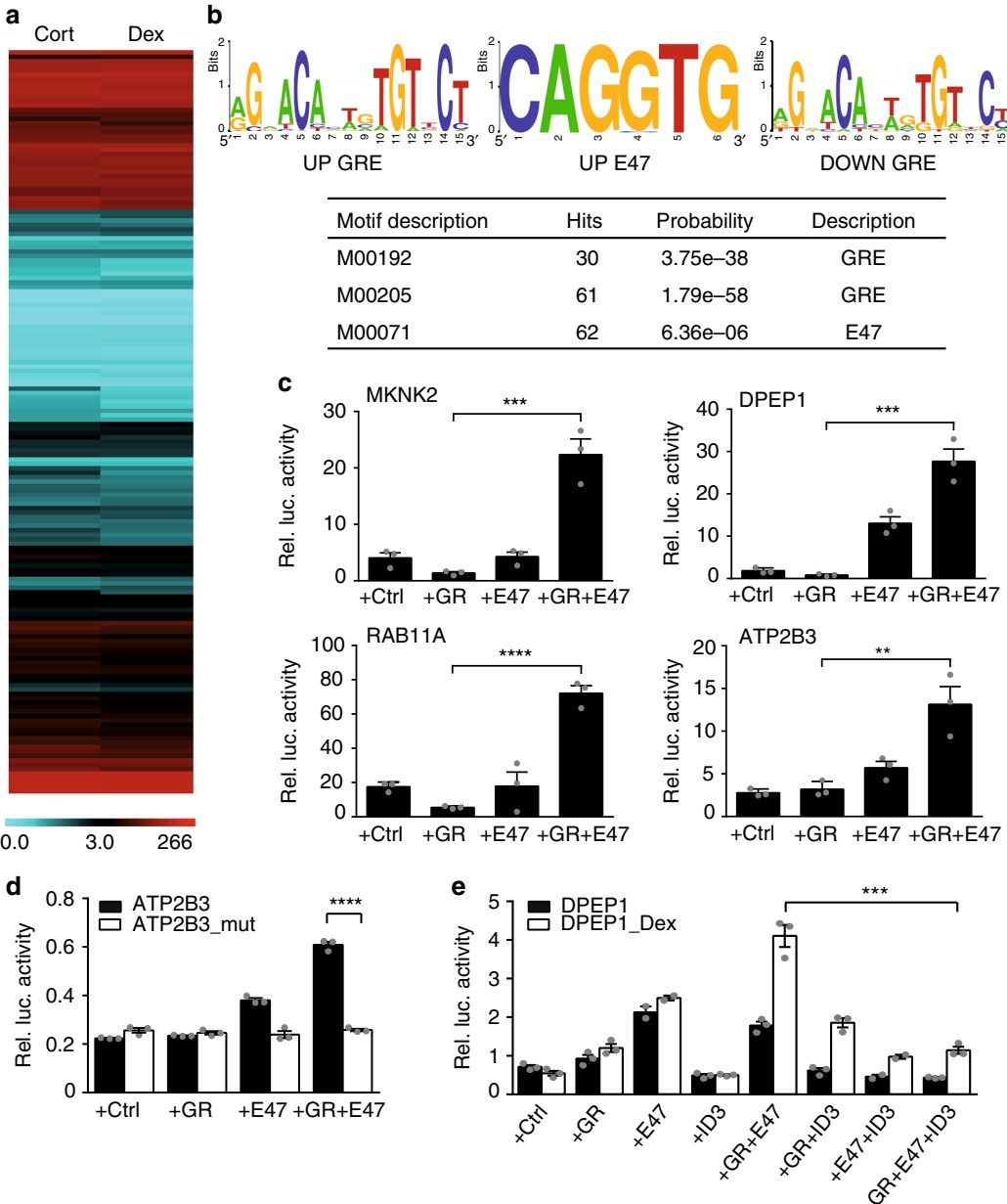

**Fig. 6** E47 and GR activate human *cis*-regulatory elements. **a** Clustering of the relative luciferase activity of > 160 human reporter constructs regulated by GR. The data from biological triplicates was normalized for transfection efficiency and to empty vector with vehicle. Reporters activated by GR plus ligand (Cort(isol) or Dex(amethasone)) appear red, repressed reporters appear blue (Color scale represents relative luciferase activity). **b** Motif analyses on corresponding DNA sequences show identical GRE consensus motifs in both activated and repressed reporters (UP & DOWN GREs). Upregulated reporter sequences are enriched for E47 consensus binding sites (UP E47). **c** Luciferase assays of selected reporters co-transfected with GR and E47 expression vectors in CV-1 cells treated with Dex. **d** The same luciferase assay performed with a mutant version of the *ATP2B3* regulatory element lacking E-Boxes. **e** Luciferase assay of the *DPEP1* reporter co-transfected with expression vectors for GR, E47 and ID3, in CV-1 cells treated with vehicle or Dex. All bar graphs are shown as mean ± SEM, **$P < 0.01$, ***$P < 0.001$, ****$P < 0.0001$, Student's *t* test, $n = 3$ replicates

Motif enrichment and read distribution analyses around GR peaks were conducted with HOMER. All scripts are deposited at github (https://github.com/FranziG/E47KO-liver). NGS data and annotated peak files can be accessed via the NCBI's Gene Expression Omnibus using the accession number GSE111526.

**ChIP coupled to mass spectrometry (ChIP-MS).** ChIP-MS was performed in livers of Dex-injected *Alb-Cre x E47flox/flox* and control littermates. ChIP was carried out as described with minor modifications. Chromatin was sonicated to an average size of 200 bp. For each biological condition, three replicates were used. After overnight immunoprecipitation with rabbit αGR (#24050–1-AP, Proteintech, 5 μg) and rabbit IgG (#2729, Cell Signaling, 5 μg), antibody-bait complexes were bound by protein G-coupled Dynabeads (Life Technologies) and washed three

times with wash buffer A (50 mM HEPES pH 7.5, 140 mM NaCl, 1% Triton), once with wash buffer B (50 mM HEPES pH 7.5, 500 mM NaCl, 1% Triton) and twice with TBS. Precipitated proteins were eluted with an on-bead digest[46]. Beads were incubated for 30 min with elution buffer 1 (2 M Urea, 50 mM Tris-HCl (pH 7.5), 2 mM DTT, 20 μg/ml trypsin) followed by a second elution with elution buffer 2 (2 M Urea, 50 mM Tris-HCl (pH 7.5), 10 mM Chloroacetamide) for 5 min. Both eluates were combined and further incubated over night at room temperature. Tryptic peptide mixtures were acidified with 1% TFA and desalted with Stage Tips containing 3 layers of C18 reverse phase material and analyzed by mass spectrometry.

Peptides were separated on 50-cm columns packed in-house with ReproSil-Pur C18-AQ 1.9μm resin (Dr. Maisch GmbH). Liquid chromatography was performed

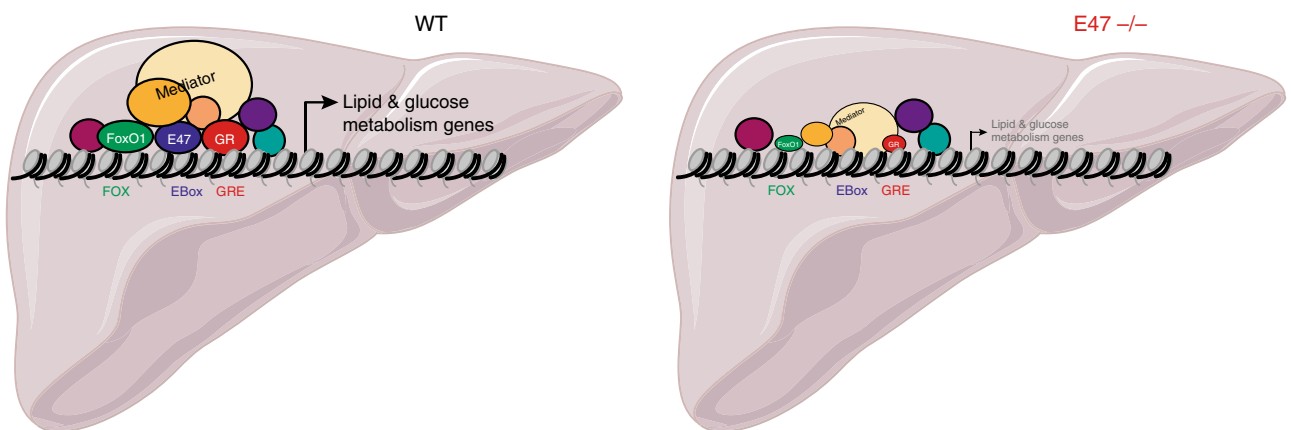

**Fig. 7** E47 modulates hepatic GR function. Graphical model summarizing our findings: In wild-type mouse livers, E47 binding at shared *cis*-regulatory elements overlapping with GR and FoxO1, Mediator recruitment leads to the transcriptional activation of a subset of metabolic genes in response to GCs. In *E47* mutant mice, reduced binding of GR and FoxO1 causes a reduction of Mediator recruitment and mRNA expression of these metabolic targets, which protects these animals from hepatic steatosis and hyperglycemia induced by GC treatment. (The liver cartoon was obtained from Servier Medical Art.)

on an EASY-nLC 1000 ultra-high-pressure system coupled through a nanoelectrospray source to a Q-Exactive HF mass spectrometer (Thermo Fisher Scientific). Peptides were loaded in buffer A (0.1% formic acid) and separated applying a non-linear gradient of 5–60% buffer B (0.1% formic acid, 80% acetonitrile) at a flow rate of 250 nl/min over 105 min. Data acquisition switched between a full scan and 15data-dependent MS/MS scans. Multiple sequencing of peptides was minimized by excluding the selected peptide candidates for 25 s. All other settings were set as previously described[47].

Raw mass spectrometry data were analyzed with MaxQuant version 1.5.3.54 and Perseus version 1.5.4.1 software packages. Peak lists were searched against the mouse UniprotFASTA database (2015_08 release) combined with 262 common contaminants by the integrated Andromeda search engine. False discovery rate was set to 1% for both peptides (minimum length of 7 amino acids) and proteins. 'Match between runs' (MBR) with a maximum time difference of 0.7 min was enabled. Relative protein amounts were determined by the MaxLFQ algorithm with a minimum ratio count of two[48]. Missing values were imputed from a normal distribution applying a width of 0.2 and a downshift of 1.8 standard deviations.

Significant outliers were defined by permutation-controlled Student's *t* test (FDR < 0.05, s0 = 1) comparing triplicate ChIP-MS samples for each antibody and biological condition.

**Co-IP and western blot**. Liver nuclear extracts were immunoprecipitated in IP buffer (20 mM Tris pH 8, 100 mM KCl, 5 mM $MgCl_2$, 0.2 mM EDTA, 20% glycerol, complete protease inhibitors) after pre-clearing using rabbit αGR antibody (#24050–1-AP, Proteintech, 3 μg) and rabbit IgG (#2729, Cell Signaling, 3 μg) for 2 h at 4 °C and then incubated with BSA-blocked α-rabbit Dynabeads (Invitrogen) overnight. Beads were washed 3x with wash buffer (20 mM Tris pH 8, 500 mM KCl, 5 mM $MgCl_2$, 0.2 mM EDTA, 20% glycerol, 1% Triton) before elution in 1 M DTT, 6x Laemmli at 37 °C for 30 min. Western blotting was performed according to standard protocols using mouse αGR antibody (#SC-393232, Santa Cruz, 1:1000) and mouse αE2A antibody (#SC-133075, Santa Cruz, 1:500).

Western blotting of nuclear extracts from Dex-treated wild type and *E47*−/− livers was performed using rabbit αGR (#12041, Cell Signaling, 1:2000) and rabbit αSnrp70 (ab83306, Abcam, 1:5000) antibodies.

**Histology**. For Oil Red O stainings on PFA-fxed liver tissue, 6μm cryosections were incubated with Oil Red O solution (#O0625, Sigma Aldrich) in isopropyl alcohol followed by hematoxylin (#HT110216, Sigma Aldrich). For H&E staining, 6 μm PFA-fixed liver paraffin sections were treated with hematoxylin and eosin Y (#GHS332, Sigma Aldrich) using standard procedures. Brightfield microscopy was performed with a Keyence BZ-9000 microscope at a magnification of ×20.

**In vivo experiments**. Dexamethasone (#D2915, Sigma Aldrich; to induce hyperglycemia) and corticosterone (#27840, Sigma Aldrich; to induce hepatic steatosis) were dissolved in ethanol and supplied in the drinking water of mice for 3 consecutive weeks at a final concentration of 10ug/ml (Dex) and 100 μg/ml (Cort)[18–20]. Glucose tolerance tests (GTT) were performed in mice fasted overnight. Glucose (20% D-glucose; Sigma Aldrich) was injected i.p. at 2 g/kg and blood glucose levels were determined using a glucometer (AccuCheck Aviva, Roche Diagnostics). Pyruvate tolerance tests (PTT) were performed in mice fasted for 5 h. Sodium pyruvate (#5280, Sigma Aldrich) was injected i.p. at 2 g/kg. Body composition was measured using quantitative nuclear magnetic resonance technology (EchoMRI, Houston, TX). For the Dex suppression test, a single dose was injected i.p. at 1 mg/kg and mice were sacrificed 6 h later. In our hands, hyperglycemia was

reproducibly induced by our Dex protocol, while hepatosteatosis was more prominent in our Cort regimen.

For fasting experiments, food was withdrawn in the late afternoon, with mice having free access to water for the course of the experiment. Blood glucose levels and body weight were measured at 0 h and at 24, 40 and 48 h. Mice were sacrificed after 48 h, and livers and plasma were harvested for further analysis.

**Isolation of bone marrow-derived macrophages**. Bone marrow derived macrophages were isolated and treated as described previously:[9] In short, bone marrow was harvested from murine hind legs. Erythrocytes were lysed (150 mM $NH_4Cl$, 10 mM $KHCO_3$, 0.2 mM EDTA). Ficoll-Paque (GE Healthcare) was used for gradient centrifugation. Differentiation medium (DMEM, 30% L929 supernatant, 20% FBS, 1% Pen/Strep) was added on bacterial plates for 5 days. Differentiated macrophages were counted and seeded in macrophage serum free medium (Thermo Fisher Scientific) with 1% Pen/Strep.

**Liver immunohistology**. 6μm PFA-fixed liver cryosections were subjected to antigen retrieval by boiling for 30 min in citrate-based buffer. Labeling with rabbit αGR (#12041, Cell Signaling, 1:200) and α-rabbit Alexa 488 IgG (#A-21206, Life technologies, 1:250) and DAPI was performed using standard protocols.

**Luciferase assay screening and data analysis**. GR responsive human promoter/enhancer luciferase reporter constructs were obtained from Switchgear Genomics (GR pathway) and transfected together with an expression vector for human GR and a CMX-LacZ plasmid for normalization. Luciferase assays were performed in CV-1 cells treated overnight with either vehicle, 100 nM Dexamethasone or 100 nM Cortisol as previously described[9]. Relative luciferase activity was compared to vehicle and empty vector and clustered based on fold changes.

Overrepresented motifs in up- and downregulated cis-regulatory sequences were identified using OTFBS based on TRANSFAC and aligned using T-Coffee; motif position weight matrices were generated using Weblogo.

Mutagenesis of the two predicted E47-binding sites for ATPB2 to TTGGCC was performed by gene synthesis (Eurofins Genomics). For Fig. 1d and e, pRL-TK Renilla was used for normalization and luminescence was measured using the Dual-Glo kit (Promega) according to manufacturer's instructions.

**Plasma and liver tissue analyses**. Cardiac blood was collected in EDTA-coated tubes and plasma was obtained after centrifugation. Corticosterone levels were measured by an enzyme immunoassay kit (#K014-H1, Arbor assays) according to the manufacturers' instructions.

For triglyceride measurements, liver tissue was first digested in ethanol / 30% potassium hydroxide (2:1, v/v) at 60 °C. Digested samples were mixed with 1 M MgCl2 at a ratio of 1.08:1 (volume) and incubated on ice for 10 min followed by centrifugation. Samples were measured by colorimetric assay (#290–63701, Wako Chemicals) following the company's protocol.

**RNA isolation, cDNA synthesis, and RT qPCR**. Total RNA from tissue and cells was isolated using the RNeasy Mini kit (Qiagen) and reverse-transcribed with the QuantiTect Reverse Transcription Kit (Qiagen) following manufacturer's instructions. RT-qPCR was performed using Power SYBR Green Master Mix (Life Technologies) in a ViiA 7 Real-Time PCR System (Thermo Fischer Scientific). Gene expression was normalized to *U36b4*. Primers are listed in Supplementary Table 1b.

**RNA- sequencing and data analysis**. Total RNA was isolated using the RNeasy minikit (Qiagen) (liver) or Qiazol followed by the RNeasy (muscle, fat). RNA quality was verified using a 2100 Bioanalyzer with RNA 6000Nano Reagents (Agilent Technologies).

Library preparation and rRNA depletion was performed using the Illumina TruSeq stranded/unstranded mRNA Library Prep Kit v2 chemistry in an automated system (Agilent Bravo liquid handling platform) starting with 1 µg total RNA for each sample. Libraries were sequenced on the Illumina HiSeq2500 or HiSeq4000. Sequencing quality was assessed with FastQC (http://www.bioinformatics.babraham.ac.uk/ projects/fastqc/). Reads were mapped to the mouse genome mm10 (Ensembl build 38.91) and reads per gene were counted using STAR version 2.4.2a[49]. Gene count normalization and differential expression analysis was performed using DESeq2[50]. For gene annotation, biomaRt was used[51]. Functional enrichment according to gene ontology was carried out using GOrilla[52]. All scripts are deposited at github (https://github.com/FranziG/E47KO-liver). Sequencing data, normalized count data and DESeq2 output can be accessed via NCBI's Gene Expression Omnibus using the accession number GSE111877.

**Statistics**. For differences between 2 groups, unpaired two-tailed Student's $t$ test was performed. Two-way ANOVA followed by Bonferroni's multiple comparison test was used to compare more than 2 groups. For GTT and PTT, the area under the curve (AUC) was calculated from the individual glucose excursion curves. Results are given as mean ± SEM unless otherwise specified. A $p$ value < 0.05 was considered significant. All tests were performed using Graph Prism 6 (GraphPad Software, San Diego, CA USA).

## Data availability

All the data supporting this article are available upon reasonable request. ChIP and RNA NGS data sets have been deposited at GEO (NCBI) [https://www.ncbi.nlm.nih.gov/geo/]: GSE111526 (ChIP-Seq); GSE111877 (RNA-Seq). The proteomics data have been deposited to the ProteomeXchange Consortium via PRIDE [https://www.ebi.ac.uk/pride/archive/] with the identifier: PXD010157. Requests for materials, reagents or correspondence should be addressed to N. H. Uhlenhaut: henriette.uhlenhaut@helmholtz-muenchen.de.

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

## Acknowledgements

We would like to sincerely thank F. Greulich, C. Gösele, A. Schiche, J.W. Jonker, H. Juguilon, Switchgear Genomics, F. Quagliarini, J. Gerdes, K. Beresowski, T. Horn, B. Haderlein, V. Sportelli, S. Schön and S. Regn for their help with this study. We are grateful to E. Graf, T. Schwarzmayr and T. Strom for their contributions to the NGS experiments. This work was supported by the German Research Foundation DFG Emmy Noether grant to NHU (UH 275/1–1), and the Fondation Leducq and the Federal Ministry of Education and Research (BMBF CaRNAtion) grants to N.H. R.M.E. is an Investigator of the Howard Hughes Medical Institute at the Salk Institute and March of Dimes Chair in Molecular and Developmental Biology.

## Author contributions

M.C.H. and N.H.U. designed and conducted experiments; R.M.E., N.H. and N.H.U. secured funding and supervised the work; M.W. performed experiments; K.S. and M.D. provided reagents and N.H.U. wrote the manuscript together with M.C.H.

## Additional information

**Competing interests:** The authors declare no competing interests.

