## [Peer Review File · Nature Communications]

Reviewers' comments:

Reviewer #1 (Remarks to the Author):

This is a very interesting manuscript and studies with novel findings. The data show a specificity of transcription factor A47 for metabolic promoters responding to glucocorticoids. There has been an ongoing search for a way to separate metabolic from immune effects of glucocorticoids and this study is a step forward. The paper is well written.

Reviewer #2 (Remarks to the Author):

E47 modulates hepatic glucocorticoid action

In the manuscript it is examined whether co-occupancy of the glucocorticoid receptor and E47 plays a role for the transcription of a subset of genes and whether E47 modulates GR-dependent gene activation in a tissue-specific manner. The authors show that crosstalk between E47 and GR plays a role in hepatic lipid and glucose metabolism. They demonstrate that loss of E47 affects GR's ability to upregulate metabolic target genes. Consequently, E47 mutant mice appeared to be protected from steroid-induced hyperglycemia, dyslipidemia and hepatic steatosis. Using ChIP approaches the authors map GR binding in mouse livers together with hepatocyte-specific E47 mutant mice, and demonstrate that GR and E47 synergize to mediate the metabolic actions of GCs at the genomic level. They find that inactivation of E47 leads to reduced occupancy of GR, Mediator and FoxO1 at a subset of hepatic enhancers and promoters. The authors confirmed the relevance of these observations for human disease in a high throughput luciferase reporter screen of human cis-regulatory sequences responding to GCs. Thus, these data suggest that targeting this mechanism might provide an entry point for the development of GC therapies with reduced metabolic side effect profiles.

This is an excellent study. The data are interesting, well described and convincing. In particular the association of E47 and the Mediator complex are of interest. Very compelling.

Comments:

1. Line 131. E47 lack mature B cells. E47 null mutant mice are blocked at the onset of B cell development. Specifically, at the pro-B cell stage.
2. Line 157. E47 frequently acts as a transcriptional repressor.

Reviewer #3 (Remarks to the Author):

The manuscript “E47 modulates hepatic glucocorticoid action” by Hemmer et al. investigates the role of E47 in regulating the hepatic stress hormone pathway. The authors find E47 is required for sufficient GR, as well as co-activator binding to metabolic loci within the liver. The authors further demonstrate mice deficient in E47 both globally as well as specifically in the liver, are protected against glucocorticoid-induced diabetes and dyslipidemia. While the manuscript is well written and well controlled, the physiological relevance of this crosstalk is lacking. Please see note below.

1. It is clear E47 promotes metabolic dysfunction in a hypercorticosteroid environment. However, glucocorticoids are secreted in response to metabolic stress such as prolonged fasting. In figure 1 the authors utilize the natural rise in circadian glucocorticoid secretion to examine the hepatic GR cisome and found E47 binding motifs enriched. This clearly indicates E47 cooperates with GR under physiological settings. Could the authors provide reasoning as to why a basal phenotype of hypoglycemia, for example, is not observed in either the global E47 KO or liver-specific KO? Furthermore, what happens to the fasting response in E47 KO mice? Are they hypoglycemic similar to the hepatocyte-specific GR KO? Furthermore, do you lose GR binding to gluconeogenic genes in response to fasting the E47 KO mouse? Addressing this point would provide more physiological insight into the interaction between GR and E47 in regulating gene expression.
2. The cartoon in figure 6 is too general. Should be more liver centric, such as “protection from hepatic side effects”.
3. The authors provide references for their models of excessive glucocorticoids, however, we believe this should be included in the material and methods for the reader.

Response to Reviews

Hemmer et al., E47 modulates hepatic glucocorticoid action

NCOMMS-18-15844

Point by point response to reviewers' comments:

First of all, we would like to sincerely thank all the reviewers for taking the time to thoroughly evaluate our manuscript, and for their constructive criticism. We believe we have now addressed all points raised, see below.

Reviewer #1 (Remarks to the Author):

This is a very interesting manuscript and studies with novel findings. The data show a specificity of transcription factor E47 for metabolic promoters responding to glucocorticoids. There has been an ongoing search for a way to separate metabolic from immune effects of glucocorticoids and this study is a step forward. The paper is well written.

Thank you!

Reviewer #2 (Remarks to the Author):

E47 modulates hepatic glucocorticoid action

In the manuscript it is examined whether co-occupancy of the glucocorticoid receptor and E47 plays a role for the transcription of a subset of genes and whether E47 modulates GR-dependent gene activation in a tissue-specific manner. The authors show that crosstalk between E47 and GR plays a role in hepatic lipid and glucose metabolism. They demonstrate that loss of E47 affects GR's ability to upregulate metabolic target genes. Consequently, E47 mutant mice appeared to be protected from steroid-induced hyperglycemia, dyslipidemia and hepatic steatosis. Using ChIP approaches the authors map GR binding in mouse livers together with hepatocyte-specific E47 mutant mice, and demonstrate that GR and E47 synergize to mediate the metabolic actions of GCs at the genomic level. They find that inactivation of E47 leads to reduced occupancy of GR, Mediator and FoxO1 at a subset of hepatic enhancers and promoters. The authors confirmed the relevance of these observations for human disease in

a high throughput luciferase reporter screen of human cis-regulatory sequences responding to GCs. Thus, these data suggest that targeting this mechanism might provide an entry point for the development of GC therapies with reduced metabolic side effect profiles.

This is an excellent study. The data are interesting, well described and convincing. In particular the association of E47 and the Mediator complex are of interest. Very compelling.

Comments:

1. Line 131. E47 lack mature B cells. E47 null mutant mice are blocked at the onset of B cell development. Specifically at the pro-B cell stage.

We agree that this statement should be more specific and have added the following words on **p.6**:

'Homozygous E47 mutant mice lack mature B cells due to a failure to progress past the pro-B cell stage and have a partial early block in T-lymphocyte development.'

2. Line 157. E47 frequently acts as a transcriptional repressor.

Thank you for pointing this out, this is a very important issue. We have changed the wording on **p.7** accordingly:

'Importantly, the majority of genes affected by the loss of E47 were downregulated, as one would anticipate from diminished GR activity. Both GR and E47 can also act as transcriptional repressors, which might explain the upregulated transcripts.'

Reviewer #3 (Remarks to the Author):

The manuscript 'E47 modulates hepatic glucocorticoid action' by Hemmer et al. investigates the role of E47 in regulating the hepatic stress hormone pathway. The authors find E47 is required for sufficient GR, as well as co-activator binding to metabolic loci within the liver. The authors further demonstrate mice deficient in E47 both globally as well as specifically in the liver, are protected against glucocorticoid-induced diabetes and dyslipidemia. While the manuscript is well written and well controlled, the physiological relevance of this crosstalk is lacking. Please see note below.

1. It is clear E47 promotes metabolic dysfunction in a hypercorticosteroid environment. However, glucocorticoids are secreted in response to metabolic stress such as prolonged

fasting. In figure 1 the authors utilize the natural rise in circadian glucocorticoid secretion to examine the hepatic GR cisome and found E47 binding motifs enriched. This clearly indicates E47 cooperates with GR under physiological settings. Could the authors provide reasoning as to why a basal phenotype of hypoglycemia, for example, is not observed in either the global E47 KO or liver-specific KO? Furthermore, what happens to the fasting response in E47 KO mice? Are they hypoglycemic similar to the hepatocyte-specific GR KO? Furthermore, do you lose GR binding to gluconeogenic genes in response to fasting the E47 KO mouse? Addressing this point would provide more physiological insight into the interaction between GR and E47 in regulating gene expression.

We thank the reviewer for raising this important point.

With respect to a potential basal phenotype: A study from the Schütz lab showed that liver-specific *GR* mutant mice do not display basal hypoglycemia either (*Opherk et al., Molecular Endocrinology 2004*). This might actually be a good thing when considering future therapeutic approaches targeting E47.

In order to study the physiological relevance of our observations, we have now fasted both *E47* knockout and liver-specific mutant mice for up to 48 hours. **New figure 5** now shows GR & Med1 binding, gene expression, blood glucose and liver triglycerides in response to fasting.

We did observe reduced expression of certain metabolic genes together with reduced occupancy of GR, Med1 and FoxO1. However, while there was a trend for lower glucose in fasted *E47* mutants, this did not reach statistical significance. It could be that this genomic reduction is not enough to lower total circulating glucose significantly in fasted mice (The *E47* mutant phenotype is reduced GC action, but not total loss of function.). In comparison, our model of chronic GC treatment with high exogenous doses might be more severe. Also, the Harris lab has shown that reduced hepatic GR gluconeogenic function can be compensated by the kidney (*Bose et al., Endocrinology 2016*). In contrast, fasted *E47* mutants indeed have lower liver triglycerides, which could be indicative of a physiological role of E47 in untreated mice.

2. The cartoon in figure 6 is too general. Should be more liver centric, such as 'protection from hepatic side effects'.

We agree. We have changed **figure 7** (formerly figure 6) accordingly, which now displays 'hepatic side effects'.

3. The authors provide references for their models of excessive glucocorticoids, however, we believe this should be included in the material and methods for the reader.

Thank you for pointing out this oversight. We have expanded the methods section accordingly and include references to *Di Dalmazi et al.*, *Karatsoreos et al.* and *Fransson et al.* (see **p.19**):

In vivo experiments

Dexamethasone (#D2915, Sigma Aldrich; to induce hyperglycemia) and corticosterone (#27840, Sigma Aldrich; to induce hepatic steatosis) were dissolved in ethanol and supplied in the drinking water of mice for 3 consecutive weeks at a final concentration of 10µg/ml (Dex) and 100 µg/ml (Cort) ^{18,19,20}.

We hope we were able to address all the points raised to the reviewers' satisfaction.

REVIEWERS' COMMENTS:

Reviewer #2 (Remarks to the Author):

This is an interesting study. The data are convincing and well presented. They are also of wide significant interest.

Reviewer #3 (Remarks to the Author):

I am satisfied with the authors responses to my original critique.

Hemmer, MC et al., E47 modulates hepatic glucocorticoid action, NCOMMS-18-15844A

Point-by-point response to REVIEWERS' COMMENTS:

Reviewer #2 (Remarks to the Author):

This is an interesting study. The data are convincing and well presented. They are also of wide significant interest.

We thank the reviewer for her/his time and efforts.

Reviewer #3 (Remarks to the Author):

I am satisfied with the authors responses to my original critique.

We thank the reviewer for taking the time to thoroughly evaluate our manuscript and for her/his helpful suggestions.